# Monitoring Tools and Strategies for Effective Electrokinetic Nanoparticle Treatment

**DOI:** 10.3390/nano13233045

**Published:** 2023-11-29

**Authors:** Huayuan Zhong, Henry E. Cardenas

**Affiliations:** College of Engineering and Science, Louisiana Tech University, Ruston, LA 71272, USA; cardenas@latech.edu

**Keywords:** Portland cement, pozzolanic, electrokinetic treatment, nanoparticle, turbidity, coagulation

## Abstract

Nanoparticles are increasingly being used by industry to enhance the outcomes of various chemical processes. In many cases, these processes involve over-dosages that compensate for particle losses. At best, these unique waste streams end up in landfills. This circumstance is inefficient and coupled with uncertain impacts on the environment. Pozzolanic nanoparticle treatments have been found to provide remarkable benefits for strength restoration and the mitigation of durability problems in ordinary Portland cement and concrete. These treatments have been accompanied by significant particle losses stemming from over-dosages and instability of the colloidal suspensions that are used to deliver these materials into the pore structure. In this study, new methods involving simple tools have been developed to monitor and sustain suspension stability. Turbidity measurement was introduced to monitor the progress of electrokinetic nanoparticle treatment. This tool made it possible to amend a given dosing strategy in real time while it remains possible to make effective treatment adjustments. By monitoring the particle stability and using pH and electric field controls to avoid suspension collapse, successful electrokinetic treatment dosage strategies were demonstrated using 20 nm NALCO 1056 alumina-coated silica particles. These trials indicated that turbidity measurements could track the visually imperceptible phenomena of particle flocking early on at the inception of its development. Suspensions of these nanoparticles were successfully delivered into 5 cm diameter by 10 cm tall hardended cement paste (HCP) specimens by monitoring fluid turbidity along with the specific gravity and using these values to guide the active management of the treatment dosage and pH. Under this new strategy, these losses were reduced to 5% as compared to the 80% losses associated with other treatment approaches. The relationship between the turbidity and the specific gravity was found to be linear. These plots indicated regions of turbidity and specific gravity that were associated with particle flocking. The tools, guidelines, and strategies developed in this work made it possible to manage efficient (low-particle-loss) electrokinetic nanoparticle treatments by signaling in real time when adjustments to electric field, pH, and particle dosage increments were needed.

## 1. Background

### 1.1. Theory of Electrokinetic Nanoparticle Treatment

Electrokinetic treatment can be utilized to transport charged species into or out of porous materials [1]. A given treatment process may exhibit several phenomena including electro-osmosis, electrophoresis, and ion migration, among others [2]. Many of these electrokinetic processes are observed when an electrical potential gradient is applied throughout a porous material that is saturated with a conductive liquid. This applied voltage tends to cause a current to flow through the fluidic pathways of the circuit. The overall current in the circuit consists of the drift of charged species (ions and colloidal particles) traveling between the electrodes that are provided for treatment. This drift current passes through the pores of the material and into the electrochemical reactions that may occur at each electrode. To achieve a successful and efficient treatment application, emphasis needs to focus on maintaining particle stability. In general, nanoparticle stability is governed by several particle and suspension fluid properties. However, electrokinetic treatment processes can change some of these parameters and thus tend to destabilize the system. 

In general, the stability of a given EN (electrokinetic nanoparticle) treatment is relatively sensitive to process parameters. Providing appropriate treatment settings for these parameters (electric field, pH, temperature, and others) would tend to enhance the effectiveness of particle transport, since the transport process is mainly governed by electrophoresis. Particle transport can also be influenced by electro-osmosis, diffusion, and pressure flows [1,3]. Regarding electrophoresis, several factors including the ionic strength of the fluid, surface potential of the particle, and dielectric constant of the fluid play important roles regarding the transport of suspended particles in an EN treatment. In this fluid suspension, the particles are surrounded by a cloud of ions attracted to an array of charges that are present on the nanoparticle surfaces. These conditions cause particles to exhibit a net charge. As the nanoparticles wander about due to Brownian motion, they tend to repel each other due to this net charge. This repulsion phenomenon is a critical stabilization mechanism that derives from the electrostatic interaction of these respective ion clouds. In theory, these ion clouds exhibit what is referred to as a double-layer structure that in turn exhibits a zeta potential, as shown in Figure 1 [4,5]. This zeta potential can be taken as a measure of how forcefully and effectively these particles repel each other. According to Derjaguin–Landau–Verwey–Overbeek (DLVO) theory, the stability of particles was found to be dictated by a balanced force system that includes attractive Van Der Waals forces and repulsive electrostatic forces associated with surface charges [6,7]. The electrostatic repulsion from the surface charges on the particles helps them remain stable and separated in the suspension [8]. If two particles come relatively close together, attractive Van der Waals forces can overcome the repulsive electrostatic forces, thus causing the particles to stick together [5,9]. If this sticking behavior becomes widespread in a given system, it can lead to the unstable collapse of the particle suspension. This phenomenon can be a significant cause of charge carrier loss and the associated rise in circuit resistance during a given treatment. This can correlate to a lower effective dose of particles reaching the target area of a given treatment.

For a given EN treatment, the applied electric field (E-field) will generate an electrical potential gradient located within and adjacent to the treatment subject, which drives the charged particles into the target material [10,11]. At the time the electrode functions as the anode in a specified treatment, the prevailing electrolysis reaction is represented as follows:2H_2_O = O_2_ + 4H^+^ +4e^−^

This reaction generates H^+^ ions (hydrogen ions), contributing to a reduction in pH in the proximity of the anode. Conversely, the typical reaction transpiring at the cathode can be expressed as shown below:2H_2_O + 2^e−^ = H_2_ + 2OH^−^

Resultantly, hydrogen gas (H_2_) and hydroxide ions (OH^−^) are produced at the cathode, leading to heightened concentrations of these species in the vicinity of the cathode. 

The benefits achieved from a given electrokinetic treatment depend upon how successfully the driven nanoparticles can enter into the porous material [12]. Ben-Moshe and other researchers observed that a notable parameter that affects the electrokinetic mobility of a nanoparticle is the ionic strength of the suspension [13,14]. The composition of the ions in the suspension fluid also influences this mobility. Together, the composition and ionic strength strongly influence the zeta potential. A relatively large zeta potential can cause a particle to exhibit high mobility. Both the mobility and stability of a particle suspension can easily be manipulated by a wide range of external factors. These factors include the driving electrical field strength used during a treatment, the pH, the particle concentration, the chemical composition of the suspension, and the temperature [15,16].

### 1.2. Particle Destabilization Mechanisms

Several studies have investigated various particle destabilization mechanisms [15,17]. Zhou and others examined suspended particle gelling behavior, which is also referred to as coagulation. The collapse of a particle suspension (noted earlier) often results in the formation of a coagulated gel. In these studies, the pH and ionic strength of the suspensions greatly influenced the nanoparticle gelling rates [18,19]. In other work, the authors observed that a high electric field strength can drive suspended particles into close mutual proximity as they approach a path bottleneck, such as pore openings on an ordinary Portland cement surface. When forced close together, they are now under a high risk of colliding and sticking together. By extension, should a large system of particles fall subject to this collision risk as they approach a porous surface, they may form an electro-coagulated gel [15]. Prior to suffering coagulation, the particles may become concentrated at a porous surface to which they are being driven (as they wait their turn to enter a pore) but remain stable for a short period of time. During this waiting period, pH changes may diminish the zeta potentials of these particles, causing them to approach each other, collide, and gel. Particles lost to a gel are considered lost since they can no longer be transported electrokinetically. 

Another study examined the behavior of polymeric particles that were configured with smaller particles adsorbed onto their surfaces, resulting in flocculation [20,21]. These flocs remained suspended. This demonstrated that particle losses caused by the flocculation may not lead directly to coagulation; however, flocked particles tend to be too large to penetrate the surface pores of the cement during a given treatment. In work conducted by Hotze and Phenrat, it was observed that the larger surfaces associated with flocked nanoparticles led to a higher tendency for collisions [22]. 

### 1.3. Turbidity of a Suspension

Turbidity analysis entails the investigation of optical phenomena leading to the scattering and absorption of light in water, deviating it from a rectilinear transmission path. Turbidity manifests as opaqueness or diminished clarity in water [23]. The direction of transmitted light is altered upon interaction with particles within the water column. The quantification of suspended particle concentration, such as silt, clay, algae, organic matter, and microorganisms, within water is facilitated by the detection of light that is scattered by these entities [23]. To detect changes in suspension characteristics, other work in hydrology and geomorphology employed turbidity measurements to quantify suspended sediment [24,25]. Numerous studies have shown that the clarity of a suspension expressed in terms of suspended sediment concentrations can be predicted by using turbidimeter measurements [26,27,28].

The International Organization for Standardization (ISO) gave the most recent definition of turbidity as ‘the reduction of transparency of a liquid caused by the presence of undissolved matter’ in 2019 [29]. Turbidity results can be impacted by the particle size, shape, and composition in addition to watercolor [30,31,32].

The forward light-scattering meter was widely used for turbidity measurement [33,34]. The principle of operation of this type of meter involves measuring the ratio of LED light, which is scattered over a range of angles with respect to forward transmitted light. These values are calibrated against the same ratios for a standard suspension of Formazine [31,32].

## 2. Methodology and Experiment Setup

The works contained in this section focused on examining the outcomes of several dosing strategies. These strategies were applied to electrokinetic nanoparticle treatments on concrete cylinders. HCP specimens were used in this study.

### 2.1. Batching and Curing

The binder used in this study consisted of low-alkali, ordinary Type I/II Portland cement (Ash Grove Cement Company, Little Rock, AR, USA) and deionized water in a 0.48 water-to-cement ratio. Portland cement is commonly used in constructing modern structures worldwide [35]. The compositions of the cement power are shown in Table 1.

The dimension of the cylindrical HCP specimens conducted in this study is illustrated in Figure 2. Several electrokinetic test treatments were set up and conducted with cylindrical specimens as shown in this figure as well. Mixed-metal-oxide-coated titanium wire was embedded in each of the specimens. This wire is 1/16 inch (1.5875 mm) in diameter. It is manufactured by Corrpro (AEGION Corp, St. Louis, MO, USA) for cathodic protection applications. To provide an electric field that was relatively uniform throughout each part of the specimen, the length of the embedded wire was limited to 2 inches (50.8 mm). This constraint provided equivalent distances between the wire and both the bottom and the side surfaces of each specimen.

The batching process complied with ASTM C 305 to make a relatively uniform performance batch [36]. A low-speed mixer (Kitchen Aid, Classic Model, Whirlpool Corporation, Greenville, Ohio) was used for batching. The specimens were demolded after 24 h and moist cured in lime water (2.5 g/L Ca(OH)_2_ solution) for 2 weeks.

The nanopozzolan particle used in this study was the NALCO 1056 colloidal suspension (NALCO Water, Bedford Park, IL, USA). It is a positively charged, 24 nm, aluminum-coated silica particle sol. NALCO 1056 has been studied by Cardenas and others, and researchers observed positive results in enhancing strength and providing chemical resistance. The silica and alumina content of this particle system exhibits pozzolanic reactivity that yields binder phases that are chemically similar to the binder material that is native to Portland cement [1,2,3,4,15,16]. The dosage was designed in terms of the volume percentage of particles available in a given treatment bath. This was conveniently managed in terms of the weight percentage (wt%) of particles available in the treatment fluid by simply monitoring the specific gravity of this fluid. Dosage values started at 0.04 wt% particle content (0.03% volume concentration). The applied electric field strength was kept under the electro-coagulation threshold value of the NALCO 1056 nanoparticles, 0.4 V/cm [15].

### 2.2. pH and Turbidity Measurement

Each treatment trial was run for up to 4 days. The daily dosage particle increments varied for different trials. The pH of the suspension was monitored and adjusted by the addition of hydrochloric acid (HCL). PH monitoring was conducted at four locations, A–D, as shown in Figure 3. All treatments were performed at standard laboratory temperature (20 °C).

Visual observation of treatment fluid appearance was recorded daily. Three other suspension stability parameters were monitored as well. The turbidity was measured via a Hach 2100p turbidimeter (Hach Co., Ltd., Loveland, CO, USA). The pH of the suspension was monitored using an OAKTON pH 11 series meter (Cole-Parmer LLC, Vernon Hills, IL, USA). A CORALIFE DEEP SIX Hydrometer (Central Garden & Pet Co., Ltd., Franklin, WI, USA) was used for rapid specific gravity monitoring.

## 3. Results and Discussion

To determine an effective treatment approach that could minimize particle losses, a series of tests were conducted using the NALCO 1056, aluminum-coated silica sol.

### 3.1. Treatment Approaches Examination

Figure 4 shows the visual and parametric indications of particle treatment progress. This figure contains “top views” of beakers with the treated HCP specimens removed from the setup (see the setup in Figure 2 for reference). The beakers show the appearance of the particle suspension fluid as it changed during a 5-day treatment. The treatment was run continuously with a 0.4 v/cm electric field. The entire particle dosage was provided on the first day. During the first 2 treatment days, the turbidity of the suspension transitioned from cloudy to gradually clear. It was observed that the Nephelometric Turbidity Unit (NTU) value of the fluid was decreasing during this treatment period from 116 to 101. The specific gravity values were also decreasing from 1.009 to 1.007. At Day 3, the rate of decrease in specific gravity was slowing down. Some evidence of particle flocking was observed in the fluid and will be examined in a later section. Meanwhile, the turbidity exhibited an increase. The visually inspected transparency on this day (Day 3) was cloudier than on Day 1.

The increased turbidity in the midst of decreasing particle content on Day 3 indicates that the interaction between light photons and particle surfaces was changed. A likely source of change was probably due to the rising pH during this period. As pH rises, the magnitude of the zeta potential would tend to decrease, which would also decrease the electrostatic repulsion among the particles. This lower repulsion enabled more particles to collide and form flocks. Photons interacting with small flocks would tend to reduce the transmission of light, since there would now be additional surfaces associated with each of these collisions.

### 3.2. Flocking Behavior Observation

As noted in the Background section, the embedded electrode of the cement specimen (see setup in Figure 2) was connected to the negative pole of the power supply to attract positively charged particles. As a result, OH^−^ ions were being produced at the cathode and then diffusing into the surrounding fluid of the beaker shown in Figure 4. This continuous production of OH^−^ ions would be sufficient to cause the pH value of the suspension to rise. On Day 3, the measured pH values were approaching the particle suspension collapse threshold of 5.5. The threat of suspension collapse comes from the negative impact that a pH shift can have on the zeta potential of the particles. A zeta potential of reduced magnitude would tend to diminish the repulsive electrostatic force that keeps particles separated. When that separation is diminished, some of the particles could approach each other (due to Brownian motion), start colliding, and then form flocks of relatively large suspended agglomerations. The combined surface charges of these flocks would have allowed them to remain suspended in the treatment fluid for a limited period. The minimum size of two flocked particles would be approximately 40 nm. This happens to be about the size of a relatively large capillary pore in HCP [37]. For this reason, even small flocks could tend to be too large for pore entry. The flocks of particles were effectively considered “lost” because they were too large to penetrate the cement pores.

The turbidity meter measured the ratio of the side-scattered light intensity to that of the forward-transmitted light intensity. With the development of particle flocks, the incident light was probably blocked more effectively because the flocks would tend to be large and more closely spaced than in the original particle suspension. This could have resulted in the increasing NTU values of the turbidity measurement observed on Days 3–5 (of Figure 4) when the specific gravity was decreasing. Based on these observations, it appears that flocculation became indicated as the turbidity started increasing rapidly during the treatment period, even as the specific gravity (and thus particle content) was declining.

This treatment associated with Figure 4 was halted on Day 5. The specific gravity showed a decline (from 1.006 to 1.005) over the last 48 h. During the treatment, the turbidity increased from 165 to 270 (with 116 being the initial value). The alphabet-lettering sheet located beneath the beaker was no longer visible on Day 5. During the course of the treatment period, the pH increased from 3.5 to 5.8. This ending value was above the pH-induced coagulation threshold (5.5 for NALCO 1056).

In general, when the suspension pH is above this threshold value, all the particles would tend to exhibit flocking that would be soon followed by coagulation (collapse). In this case, the treatment was halted on Day 5, since there was significant evidence of severe flocking that would tend to prevent successful pore penetration. Because the pH of the suspension was above the threshold value, a negative impact on the zeta potential of the particles was expected to be significant. A drop in zeta potential would tend to allow an increase in particle flocking, both in terms of occurrence and flock size. As noted earlier, the reduced zeta potential would tend to cause these “large” flocks to be more closely spaced and thus more effective in blocking light transmission. It would not be surprising if these trends would tend to cause turbidity to increase significantly due to flock formation and growth. The large turbidity increase observed from Day 3 to Day 5 (in Figure 4) appeared to support this notion. It was thus evident that preventing a rise in pH and thus flocking would be expected to benefit the efficiency of particle transport. To achieve an efficient treatment and avoid particle loss due to flocking or coagulation, pH adjustments appear to be necessary to support the stability and effectiveness of a given electrokinetic nanoparticle treatment.

### 3.3. pH Control Approaches

Similar to Figure 4, the observations of Figure 5 show measurements of suspension stability involving the same nanoparticle (NALCO 1056). This treatment was also conducted with the same threshold electric field strength (0.4 V/cm) and the same single dosage that was applied initially. The only difference in this case was that active pH control was applied. During the treatment period, the transparency of the suspension ranged from cloudy to nearly clear, and the turbidity value of the fluid decreased over this period from 116 to 41 NTU. The specific gravity values also decreased from 1.009 to 1.004 during this period. Following the treatment, visual examination of the suspension fluid and the specimen (not shown) indicated that no coagulation or flocking of particles had occurred.

**Figure 5 nanomaterials-13-03045-f005:**
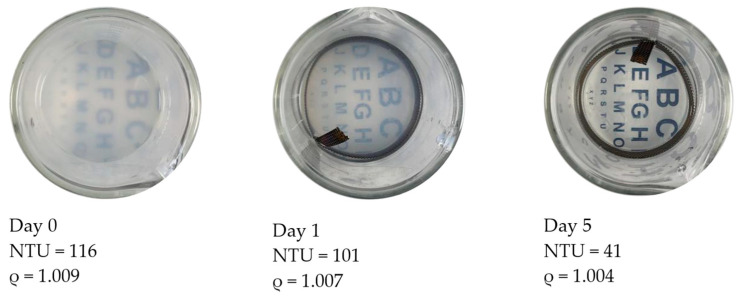
Each of these three images shows a top view of a treatment beaker after the cement specimen was removed. The sequence shows how turbidity, measured in NTUs, changed as the treatment delivered nanoparticles that were driven by a field of 0.4 V/cm. The particles used here were 24 nm, alumina-coated silica sol (NALCO1056). ρ is the specific gravity of the suspension fluid. No flocking of particles was observed during the 5-day, pH-controlled, treatment period.

With the pH adjustment applied, the electric field apparently drove stable, suspended particles that were apparently penetrating the pore openings. This was evident from declining values for turbidity and specific gravity, which indicated that these suspension particles left the treatment fluid and entered the HCP as expected. Since there were no unstable (flocked or coagulated) particles observed throughout the treatment period, this delivery process was considered efficient and successful, as the particle delivery rate (based on declining specific gravity values) increased by 25%. Based on these observations, it is evident that an effective and efficient treatment that exhibits successful particle transport into cement pores can be identified by the absence of particle loss (via flocking and coagulation) as well as declining specific gravity and turbidity values.

### 3.4. Specific Gravity Monitoring and Comparison

Figure 6 indicates that during the first 2 days of the treatment, the specific gravity dropped in value from 1.009 to 1.005. A slower rate of decrease was observed during the remaining 3 days. During the overall treatment period, the pH was monitored. Active pH control became necessary at Day 3 because the pH value of the suspension was approaching the particle collapse threshold value of 5.5. HCL was used to adjust the pH back to the starting value of 3.5.

Since OH^−^ ions were being produced at the cathode (within the HCP specimen), it caused a rise in the pH of the suspension during the first 2 days. As shown in the previous case (Figure 4), an uncontrolled pH would be expected to keep rising beyond the particle suspension collapse threshold (5.5) within a matter of days. To prevent this negative impact on suspension stability, active pH adjustment was required to stabilize the zeta potential and thus preserve the electrostatic repulsion needed to sustain the particle suspension. The relationship between the pH, the zeta potential, and the particle coagulating behavior has been well studied and established by Xiaoying and others [38]. Since the pH adjustment maintained the suspension stability, the treatment progressed well for the remaining 3 days and ultimately exhibited a relatively clear fluid. This fluid exhibited the lowest specific gravity observed (1.004) during this treatment; see Figure 5. Based on these observations, it appears that early pH adjustment could prevent treatment suspension instability by delaying the pH rise that can cause flocking problems and suspension collapse.

As shown in Figure 6, after Day 3, the specific gravity of the suspensions stopped changing significantly. This trial was halted after the fifth treatment day, since the specific gravity and the turbidity values appeared to stop responding to continued treatment. Possible reasons for this behavior will be explored in a later section. For the pH-adjusted case, the fluid looked nearly clear on the last day of treatment. At the time that these trials were stopped, the turbidity values indicated that some particles had remained in the suspension as opposed to entering the HCP pores. The final specific gravity value of 1.004 indicates that the remaining particle concentration after Day 5 was less than 50% of the starting value at Day 0.

At the end of this treatment, it was apparent that visual clarity inspection was not definitive for indicating treatment completion, since nearly 50% of the particles were still in the fluid that appeared to be clear. The visual difference between two distinct (yet relatively low) particle concentrations was barely distinguishable visually. Using such a low resolution, the visual criterion could cause residual particles to remain unutilized after the treatment has ceased. In contrast, turbidity measurements such as the NTU values of 101 and 41 from Figure 5 exhibited a significant distinction between these otherwise similar-looking fluids. These NTU numbers clearly indicated that a significant proportion of the particles had not yet entered the HCP pores. This notable difference was further confirmed by the specific gravity values of 1.007 and 1.004 that were also observed for these respective cases. Based on these observations, it appears that while a visual inspection is a convenient way for assessing the progress of transport, it is recommended that including turbidity measurement could more definitively express important suspension content changes and the associated treatment progress.

As noted earlier, it was evident visually and confirmed by turbidity and specific gravity measurements (Figure 5) that not all particles were entering the pores over a 5-day treatment. The existence of particles remaining in the treatment fluid could be due to the low electric field being used. Since the particle concentration was reduced by half after the 2nd day of treatment, the relatively low electric field might not be strong enough to force the remaining particles toward the specimen. Suspended nanoparticles tend to wander randomly due to Brownian motion. Given the relatively low particle concentration and the low field, the associated chemical and electrical gradients may not have been sufficient to overcome the Brownian motion as needed to impose a net drift into the HCP pores.

Another possible reason may explain this limited transport. At the time these treatments started, the particle concentrations inside versus outside of the HCP specimens were significantly different. The initial chemical gradient alone would have been sufficient to support transport into the HCP. As the treatment progressed, particle concentrations on each side of the specimen surface would have tended to balance. Eventually, the particle concentration inside the specimen would have become greater than on the outside. At that point, the chemical concentration gradient would then have been working against the electrical gradient. Under these conditions, the particle penetration rate would likely have been slowing down or possibly stopping, as evident in Figure 6. Meanwhile, some degree of pore openings could have been irreversibly occupied by particles during the treatment. These blockages would not have permitted particle transport in either direction. Blockages occurring in the early stages of treatment would tend to cause fewer open pores to be available, and more particles would need to wait to penetrate the pores. These conditions could have been delaying entry, increasing the population of the dense cloud of delayed particles and thus increasing the chances of instability at the HCP surface. These circumstances may explain why these remaining particles were not able to successfully penetrate the specimen surface pore openings. In future work, it is recommended that treatments be dosed with the same particle concentration on each day to overcome the development of disruptive chemical gradients.

The preceding recommendation leaves in place the fact that any given treatment will conclude with particles left behind in the treatment fluid. It is conceivable that some beneficial use could be obtained from these leftover particles. To unitize these remaining particles expediently and to further benefit the treatment results, conducting an induced electro-coagulation may be an appropriate option. Applying a relatively high electric field at the end of the treatment would tend to produce a relatively dense, coagulated particle skin on the specimen surface. In both laboratory and field tests, this skin was found to be difficult to remove and exhibited the capacity to significantly reduce the HCP surface permeability and increase the surface hardness. It is recommended that as an additional cost-efficiency measure, an electric-field-induced coagulation should be considered to utilize the leftover treatment particles and to maximize the benefit to the HCP surface properties.

### 3.5. Flocking Behavior Plot of NALCO 1056 Particles

As noted earlier, the development of increased turbidity during treatment can indicate a flocking problem. It is important to be able to detect this problem before it advances to a costly extent. With this concern in mind, Figure 7 was plotted to determine the region where the flocking behavior would be expected to appear. To investigate the transition from a stable suspension to a flocked suspension, the original particle colloid (NALCO 1056) was diluted into various particle concentrations. The key parameters plotted for this array of diluted suspensions were the specific gravity and the turbidity. As shown in Figure 7, broad range dilutions of the NALCO 1056 particle suspension were created and analyzed. The region of anticipated flocking behavior is located above the trend line that correlates the specific gravity to the turbidity.

In Figure 7, it was observed that the turbidity of the suspension tended to increase as the specific gravity increased. This makes sense, because increasing the number of particles would be expected to inhibit the transmission of light progressively. The dashed line representing the linear regression fit for the data is shown in Figure 7. The R^2^ value of 0.96 indicated a good correlation between the specific gravity and the turbidity with a 99% confidence interval. Based on these findings, the relationship between the turbidity and the specific gravity for this particle suspension was approximately linear.

The flocking region identified in Figure 7 is the range in which the particles would tend to flock. The gap in between the flocking region and the trend line can be referred to as a confidence interval gap. This gap was determined by the size of the error bars calculated for the data set. These error bars represent the uncertainty of the expected value corresponding to a 90% confidence interval for each turbidity measurement. Five trials were used to establish the mean value of each measurement. The calculation of the uncertainty of the expected value involves utilizing the mean and the standard deviation of each measurement as follows [39].
(1)$$Expected\;Value=Mean\;Value\;\pm \frac{1.654\times Standard\;Deviation}{\sqrt{number\;of\;test\;trails}}$$

This plot could provide a convenient means of assessing the stability of an ongoing treatment. For example, the treatment case presented in Figure 4 may be considered. If one were to plot the Day 1 data (NTU = 101, ρ = 1.007) onto Figure 7, this point would be located in the confidence interval gap, just below the flocking region. This gap represents a region in which the treatment suspension is potentially unstable. In contrast, the visual observation on this day in Figure 4 was that of a clear treatment fluid that appeared stable. Since the figure indicates the potential for instability; this could be a good point to intervene by adjusting the pH of the suspension to avoid flocking or particle losses that were not yet visually evident. For Day 3, the parameter data (NTU = 165, ρ = 1.006) would plot to a point that is within the bottom edge of the flocking region. The flocking behavior had clearly appeared visually by this time (as shown in Figure 4) but was not relatively serious. This visual inspection indicated a slightly increased cloudiness as compared to Day 1. At this time, even though the flocking threshold had been crossed, some adjustment to the pH would still have been possible. This would have stopped the flocking trend and thus reduced the potential for additional particle loss. The Day 5 data of Figure 4 (NTU = 270, ρ = 1.005) plots to the middle of the flocking region (in Figure 7), which would predict flocking. This correlates as expected to the dense cloudy fluid exhibited in Figure 4. As mentioned earlier, the size of flocking particle pairs (44 nm) was approximately about the size of a relatively large capillary pore opening (50 nm). At this time, these flocked particles were presumably lost, since they would have been too large to enter most of the HCP pores. These findings show that identifying the flocking region of a given particle suspension may provide a convenient benchmark for assessing the risk of particle loss during a given treatment.

### 3.6. Flocking Behavior Plot of Grace CL Particles

Another commercially available nanoparticle suspension was evaluated as a potential treatment candidate. This relatively low-cost, positively charged, 20 nm particle (manufactured by W.R. Grace) was evaluated following the same procedures as the previous case (NALCO 1056 of Figure 7). The trade name for this alternative suspension is Grace CL. The relationship between turbidity and specific gravity for Grace CL was determined as shown in Figure 8. The lightly shaded region of expected flocking behavior is located above the trend line that correlates the specific gravity to the turbidity.

As shown in Figure 8, the turbidity of the suspension increased as the specific gravity increased. The R^2^ of 0.95 in this case revealed that the relationship between turbidity and specific gravity was linear. This linear trend as well as the location of the flocking region, positioned above the trend line, was similar to the behavior of the NALCO 1056 particles (see Figure 7). The error bars were calculated under the same equation, as shown in Section 3.5 for Figure 7, utilizing a coefficient applied to the standard deviation of the turbidity measurement for a 90% confidence interval. Based on these measurements and comparisons, the relationship between the specific gravity and the turbidity was approximately linear for the Grace CL (silica) particles and thus similar in nature to the NALCO 1056 (alumina-coated silica) particles.

## 4. Conclusions and Discussion

In this study, it appears that particle flocking became evident when turbidity started increasing rapidly during a given treatment period while the specific gravity (and thus the particle content) was actually declining.

(1)To achieve an efficient treatment and avoid particle loss due to flocking or coagulation, pH adjustments appear to be necessary to support the stability and efficiency of a given EN treatment.(2)The effective and efficient treatments obtained in this work exhibited successful particle transport into cement pores, which was identified by declining specific gravities and turbidities while the treatment particles remained in stable suspension.(3)This work confirmed that periodically adjusting the pH of a particle suspension back to the starting level (during a long-term treatment period) may prevent treatment suspension instability by delaying the pH rise that can cause flocking and suspension collapse.(4)While visual inspection is a convenient way for assessing particle transport progress, it is recommended that utilizing turbidity measurements could more definitively identify important particle suspension changes that can confirm acceptable treatment progress.(5)Identifying the flocking region of a given particle suspension may provide a convenient benchmark for assessing the risk of particle loss during a given treatment.(6)The relationship between the specific gravity and the turbidity was approximately linear for the Grace CL (silica) particles and thus similar to the NALCO 1056 (alumina-coated silica) particles.

## Figures and Tables

**Figure 1 nanomaterials-13-03045-f001:**
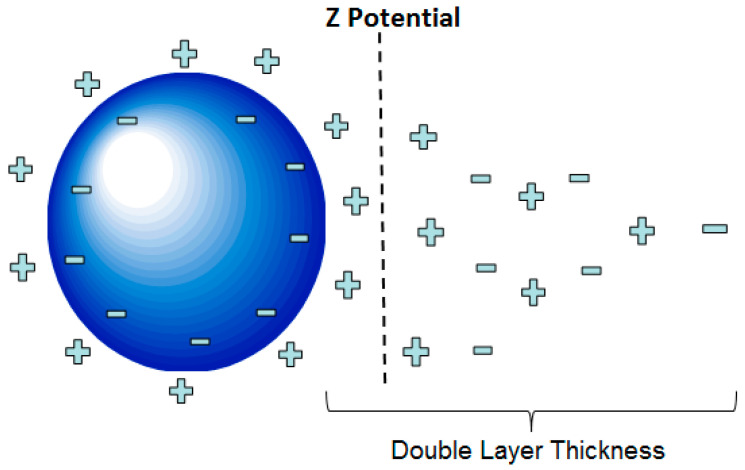
Illustration of the double-layer structure and zeta potential of a given particle.

**Figure 2 nanomaterials-13-03045-f002:**
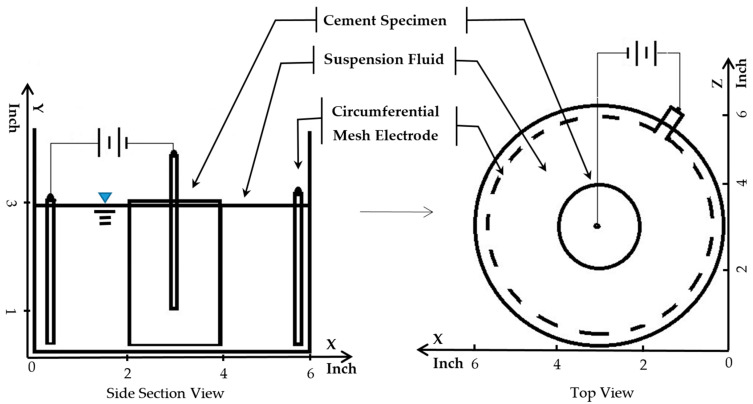
Treatment circuit setup for electrokinetic nanoparticle treatment of hardened cement paste cylinder specimens. The cylinders were 3 inches (76.2 mm) tall by 2 inches in diameter (50.8 mm).

**Figure 3 nanomaterials-13-03045-f003:**
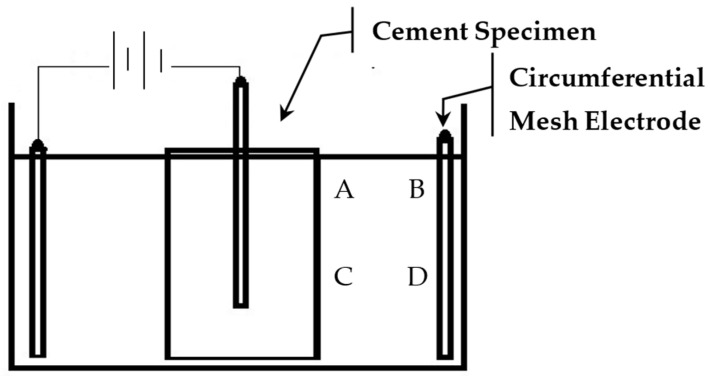
pH monitoring locations, A–D.

**Figure 4 nanomaterials-13-03045-f004:**
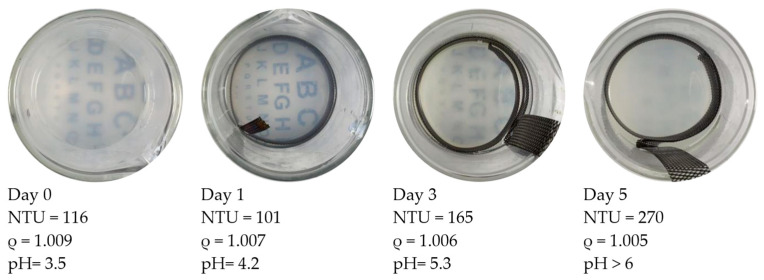
Each of these four images shows the development of pH-induced particle flocking during treatment. The sequence involved the same voltage and particle dosage as the case shown in Figure 5 but without pH control. ρ is the specific gravity of the suspension fluid. In this case, the turbidity was rising while the specific gravity was dropping during treatment. Both flocking and gelling became increasingly evident as the 5-day treatment progressed.

**Figure 6 nanomaterials-13-03045-f006:**
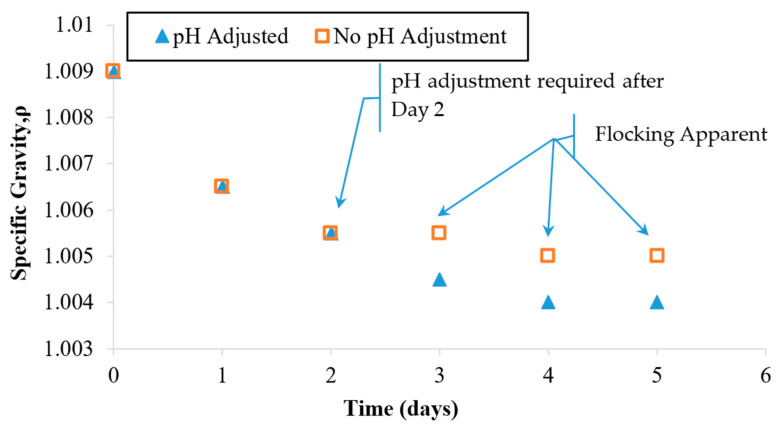
Specific gravity observations for a single dosage EN treatment applied under a controlled electric field and pH. The field was maintained at 0.4 V/cm. The pH was maintained in the range of 3.5–4.8. A parallel case was run with no pH adjustment (see Figure 4 for pH values).

**Figure 7 nanomaterials-13-03045-f007:**
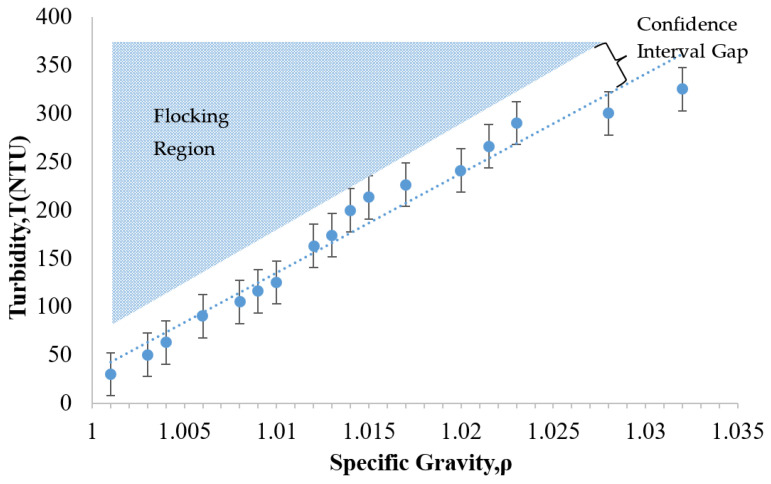
The relationship between turbidity-specific gravity and flocking behavior of 24 nm alumina-coated silica sol (NALCO 1056, NALCO Water, Bedford Park, IL, USA).

**Figure 8 nanomaterials-13-03045-f008:**
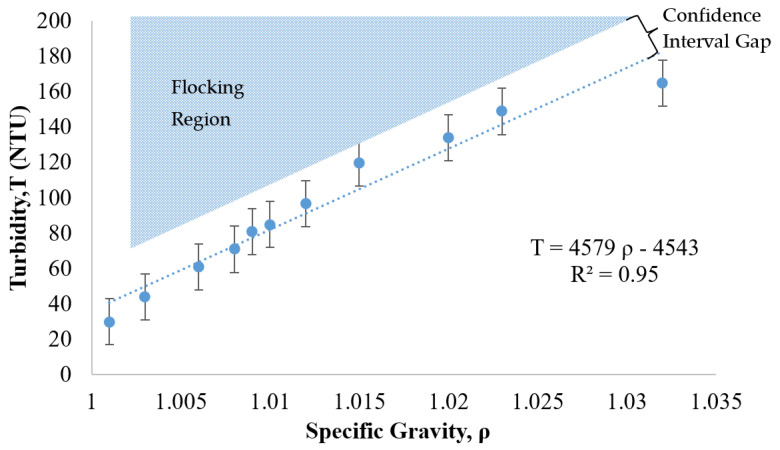
Relationship between turbidity, specific gravity, and flocking behavior of 12 nm collide silica sol (GRACE CL, GRACE, Columbia, MD, USA).

**Table 1 nanomaterials-13-03045-t001:** Mill test result of Type I/II (low alkali) cement powder used in this study *.

Component	CaCO_3_	SiO_2_	Al_2_O_3_	Fe_2_O_3_	CaO	SO_3_	Na_2_O	K_2_O
Amount (mass %)	2.41	20.15	4.62	4.03	63.61	3.20	0.16	0.57

* Cement manufactured by Ash Grove Cement Company, Little Rock, AR, USA.

## Data Availability

Data are contained within the article.

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
