# Peer review of "Monitoring Tools and Strategies for Effective Electrokinetic Nanoparticle Treatment"

_nanomaterials, 2023, doi:10.3390/nano13233045_

Round 1
Reviewer 1 Report
Comments and Suggestions for Authors
This paper presents a new method for monitoring the process of electrokinetic nanoparticle treatment by turbidity measurement and pH adjustment. Technically speaking, the main result is sound and the discussion is thoughtful. However, a major revision is recommended as follows:
1. One sketch is suggested to add in the background part to visualize the DLVO theory and the colloid dynamics.
2. Why NALCO 1056 particles and Portland cement were used in the experimental setup, please explain the motivation and the potential applications of this project.
3. Page 3 line 54, “EN” should be explained as its first present in this article.
4. Please provide more information about the mesh electrode, and if there are any electrolysis or other electrochemical reactions at the electrode?
5. Please provide more information about the specific gravity? For example, the experiment was conducted in a beaker, I am wondering how to avoid the evaporation of solutions.
6. The authors are suggested to measure the zeta potential of particles under different pH conditions to verify their speculation or explanation of flocking behavior.
Comments on the Quality of English LanguageMinor editing of English language required
Reviewer 2 Report
Comments and Suggestions for Authors
A file has been uploaded with some comments and recommendations.

Moderate English Language review should be performed to the text.
Round 2
Reviewer 1 Report
Comments and Suggestions for Authors
The authors are suggested to provide a formal "response letter" not a correction note.
Reviewer 2 Report
Comments and Suggestions for Authors
A file with suggestions has been uploaded. Also, some answer to the authors' answers is provided.

Some text reading and fine tuning of English text should be done, prior to publication.
